# Pain Catastrophising Predicts Alcohol Hangover Severity and Symptoms

**DOI:** 10.3390/jcm9010280

**Published:** 2020-01-20

**Authors:** Sam Royle, Lauren Owen, David Roberts, Lynne Marrow

**Affiliations:** Department of Psychology, School of Health and Society, University of Salford, Frederick Road, Salford M6 6PU, UK; L.J.Owen2@salford.ac.uk (L.O.); D.J.Roberts@salford.ac.uk (D.R.); L.Marrow@salford.ac.uk (L.M.)

**Keywords:** hangover, catastrophising, alcohol, veisalgia, acute hangover scale

## Abstract

Alcohol hangover is a cause of considerable social and economic burden. Identification of predictors of alcohol hangover severity have the potential to contribute to reductions in costs associated with both absenteeism/presenteeism and health care. Pain catastrophising (PC) is the tendency to ruminate and describe a pain experience in more exaggerated terms. The current study examines the possibility that this cognitive coping strategy may influence experience of alcohol hangover. The aims of the current study were to (1) examine the relationship between hangover severity and PC, (2) explore and identify discreet factors within the Acute Hangover Scale (AHS) and (3) explore whether independent factors/dimensions of acute hangover are differentially predicted by PC. A retrospective survey (*n* = 86) was conducted in which participants completed the Acute Hangover Scale (AHS); the Pain Catastrophising Scale (PCS); a questionnaire pertaining to the amount of alcohol consumed; and a demographic information questionnaire. Regression analyses showed a significant relationship between PC and hangover severity scores and demonstrated that PC was, in fact, a stronger predictor of perceived hangover severity than estimated peak blood alcohol concentrations (eBACs). Factor analysis of the AHS scale, resulted in the identification of two distinct symptom dimensions; ‘Headache and thirst’, and ‘Gastric and cardiovascular’ symptoms. Regression analyses showed that both eBAC and PCS score were significantly associated with ‘Headache and thirst’. However, only PCS score was associated with ‘Gastric and cardiovascular’ symptoms. These novel findings implicate a role for cognitive coping strategies in self-reports of alcohol hangover severity, and may have implications for understanding behavioural response to hangover, as well as suggesting that hangover and PC may be important factors mediating the motivation to drink and/or abuse alcohol, with potential implications in addiction research. Furthermore, these findings suggest that distinct alcohol hangover symptoms may be associated with different mechanisms underlying the experience of alcohol hangover.

## 1. Introduction

### 1.1. Alcohol Hangover, Symptoms and Economic Burden

Alcohol hangover is a phenomenon that occurs the day after the ingestion of alcohol, once the blood alcohol concentration (BAC) is approaching nil [1], and it is associated with a wide variety of symptoms, such as headache, nausea, and concentration problems [2,3]. Hangover is thought to be a considerable cause of economic loss through workplace absenteeism and lost productivity [4]. Researchers have also speculated that the severity of alcohol hangover is linked to the development of alcohol use disorders (AUDs) [5,6], indicating that a better understanding of the individual hangover experience and its mediators may offset the associated financial and social burden of AUD.

A number of explanations for the variance seen in alcohol hangover presentation have been suggested, including gene associations of alcohol metabolism [7,8], gender differences [9] inflammatory response to alcohol consumption [10,11], immunological functioning [12], and congener content of alcoholic drinks [13], as well as individual differences in psychosocial factors such as anxiety and mood [14], or guilt related to the actions carried out whilst drinking [3]. There is, however, little consensus regarding the biological mechanisms that underpin the experience of alcohol hangover [4], and this is particularly true for psycho-social variables. Identification of mediating factors of alcohol hangover severity may thus inform mechanistic investigations of hangover, as well as having the potential to reduce costs associated with absenteeism/presenteeism and improve health care outcomes.

### 1.2. Hangover and Risk of Alcohol Abuse

Despite the lack of mechanistic explanations for the influence of predictor variables on the experience of hangover, and mixed findings regarding relationships between familial risk for addiction and experience of hangover [15,16], there is some evidence that alcohol hangover experience may be a potential risk factor for alcohol use disorder (AUD) [15]. In this regard, hangover has been conceptualised as affecting cognitive control processes that influence local drinking behaviour. Evidence suggests that people who experience a more severe hangover will drink less, when they engage in drinking the day of a hangover [17], and that hangover can increase the time before the next alcoholic drink is consumed in frequent drinkers [18], with hangover occurrence predicting a 6 hour delay to next drink when used as sole predictor in a survival model. It is notable, however, that hangover occurrence was only associated with a delayed time to next drink in multivariate models when interacting with the onset of financial stressors, or the presence of high levels of craving at the end of the drinking episode (pre-hangover). This may implicate a role for the hangover in the delay of further engagement with drinking, when experienced alongside a continued desire/motivation to drink. The investigation of differences in factors related to motivational and inhibitive processes, such as cognitive coping strategies, during hangover, therefore has the potential to contribute to understanding of possible relationships between the hangover experience and propensity for development of AUDs.

### 1.3. Alcohol Hangover and Pain Catastrophising

Pain catastrophising (PC) has been broadly defined as an exaggerated negative orientation towards actual or anticipated pain experiences [19] and has been described as the tendency to recall pain experiences in more exaggerated terms, to feel helpless and ruminate over painful events. PC appears to be moderated, to some degree, by gender [20], psychosocial and dispositional factors [19,20]. However, despite these moderating factors, PC contributes unique and significant variance to the prediction of self-reported pain intensity, as well as to neural processing of pain [21]. Evidence has shown that the relationship between PC and pain ratings is partially mediated by diminished diffuse noxious inhibitory controls (a measure of endogenous pain inhibition), indicating a disruption in pain inhibition and suggesting a relationship between PC and pain inhibition [22]. Neurological evidence (utilising functional magnetic resonance imaging) has demonstrated that PC predicts the experience of pain, in that, during exposure to a painful stimulus, pain specific response activation in the dorsolateral prefrontal cortex (dlPFC) and medial prefrontal cortex (mPFC) correlate negatively with PC [23]. The effect of PC on brain activity in the mPFC and dlPFC also seems to be mediated by the severity of pain experienced, with reduced activity during more intense pain [24]. Additionally, the dlPFC shows greater bilateral activation during response inhibition, in comparison to interference monitoring and suppression [25], indicating some anatomical overlap between inhibitive processes and PC. It has been argued that PC may heighten pain experiences by reducing the efficiency of inhibitory pathways, though evidence for this position is indirect [26].

Alcohol hangover is characterised by pain symptoms. Indeed, the medical term for alcohol hangover “veisalgia” comes from the Norwegian kveis, which refers to the uneasiness following debauchery, and algia, the Greek term for pain. Cytokines, proteins produced during immune response that are involved in both the initiation and persistence of pathologic pain [27], are altered during hangover. Interleukin (IL-2; IL-10) and interferon (IFN-γ) cytokines have been shown to be elevated in blood during hangover [10]. In saliva, elevations of IL-2, IL-4, IL-5, IL-6, IL-10, IFN-γ, and TNF-α have been observed during hangover, and in urine, elevations of IL-4 and IL-6, as well as decreases in IL-8 have been observed in comparison to non-hangover-days [28]. These differences in cytokine levels between hangover and non-hangover days do not, however, appear to explain variance in the experience of hangover, with similar changes observed in both those reporting hangover, and those reporting hangover resistance [29]. Headache, a continuous pain in the head which has also been associated with changes in cytokine levels [29,30], represents the 3rd most common symptom of alcohol hangover [31] and symptomatic ratings of headache severity have large statistical effects in measures of hangover severity [32]. PC therefore presents a good candidate for potentially explaining some of the variance in self-reported hangover severity scores. Consequently, the current study hypothesises that greater PC will be associated with elevated hangover severity scores.

It has also been argued that hangover lacks in mechanistic explanations [4], despite the wide variety of symptoms associated with the hangover experience [2]. Certainly, dehydration is thought to represent one potential mechanism, with thirst being one of the most commonly reported hangover symptoms [2,31], due to the diuretic effects of alcohol [33]. Vasopressin levels, a biological marker of dehydration, do not, however, necessarily correlate with overall hangover severity [12]. It is possible that this is due to dehydration representing a mechanism of hangover that explains only a particular symptom cluster. Factor analysis of measures of hangover severity may therefore provide some direction for investigations of the mechanisms that give rise to symptom clusters. Furthermore, certain symptom clusters may be independently moderated by PC and may represent better predictors of alcohol abuse risk.

The current retrospective survey was therefore designed with three main aims: (i) to examine whether increased PC scores are associated with elevated hangover severity scores, (ii) to explore the factor structure of the acute hangover scale (AHS), and (iii) to explore whether different dimensions of the AHS are independently associated with PC.

## 2. Materials and Methods

### 2.1. Participants

Participants were recruited through opportunity sampling—both at a university in the north-west United Kingdom, and online via social media. Ninety-one participants completed the survey, with 3 participants excluded from analysis because they reported a non-binary gender, which presents issues for calculating blood alcohol concentrations. A further 2 participants were excluded from analyses based on age, with one reporting an age of 5 years, violating exclusion criteria, and one reporting an age of 80 years, representing an extreme outlier in the current sample (+6.24 SDs from the mean). The remaining sample of 86 had an overall mean age of 25.93 years (SD: 6.03, range: 18–46), with 51 female respondents (59%). No incentive was provided for participation. Exclusion criteria for the study were: below 18 years of age (not of legal drinking age for the area in which they reside); having not experienced hangover in the last 6 months. Exclusion was based on self-report: independent verification of these criteria was not possible with an online survey. Ethnicity was not analysed as a variable as the majority of participants (88%) self-identified as white.

### 2.2. Materials

Participants were required to complete an online survey that included the Acute Aangover Scale (AHS) [32], to measure hangover severity retrospectively, and the Pain Catastrophising Scale (PCS) [19], to measure PC associated with the experience. The PCS consists of 13 items addressing the experience of catastrophic thoughts related to the experience of pain, rated on a scale from 0 (not at all) to 4 (all the time), with total score calculated as the sum of all items. The PCS has shown a high level of reliability, with Cronbach’s α scores typically exceeding 0.8 [19,30,31], and the PCS has been validated in a variety of samples, including those ‘seeking treatment’ (vs. not seeking treatment) [34], pain outpatients (vs. community participants) [35], those with back pain (Norwegian version) [36], and those with chronic pain (Korean and Brazilian-Portuguese versions) [37,38]. The AHS consists of 9 items addressing the severity of 8 hangover symptoms, plus an overall hangover severity rating, all given on a scale from 0 (None) to 7 (Incapacitating), with AHS total score calculated as the mean of all items. The AHS has been validated for concurrent hangover severity measurement, but also shows very high correlations with scales designed for measurement of the most recent (retrospective) hangover experience [39], as well as having been utilised for recent hangover severity measurement in other research [40]. Participants reported the drinking that led to their most recent hangover using the items from McKinney and Coyle’s investigation, which asks about the number of a variety of standardised drinks consumed (e.g., pints of beer, bottles of beer, alcopops, etc.), and allows for the number of units of ethanol consumed to be calculated [41]. Demographics were also collected, including age (which was recorded for use as a covariable since previous evidence has suggested a role in the experience of hangover [42]), height and weight, (for the calculation of estimated blood alcohol concentrations), and ethnicity. Estimated blood alcohol concentrations were calculated using the method from Siedl et al. [9], resulting in the estimated peak blood alcohol concentration assuming no elimination (eBAC), and assuming a 15% elimination rate (eBAC15%). Both the AHS and PCS were adapted to reference how the participants felt during their most recent hangover.

### 2.3. Procedure

Participants, recruited via social media (Twitter, Reddit) and posters located around a university in the north-west, UK, completed an online survey hosted on Online Surveys by Jisc (https://www.onlinesurveys.ac.uk/). Participants completed the survey during their own time. The total time to complete the study was approximately 15 min. No incentives were offered for participation.

### 2.4. Ethics

The materials and methods utilised in this procedure were approved by the University of Salford Health Sciences Research ethics board (HSR1617-15), and all participants provided informed consent. The use of Online Surveys for data collection allowed for participant anonymity, and the system adheres to high ethical standards (e.g., no use of ‘cookies’ which store files to the local PC used to complete the survey).

## 3. Results

All analyses were carried out in IBM SPSS 25.0.0.1 (IBM Corp., Armonk, NY, USA).

### 3.1. Factor Analysis

A dimension reduction procedure was carried out on responses to the items of the AHS to establish whether symptom clusters existed. The item ‘hangover’ was excluded from this analysis, as this is non-symptomatic and thought to capture a broad rating of hangover severity [32]. Descriptive statistics for the remaining items are presented in Table 1. Inter-item correlations are presented in Table 2.

As all items failed to meet at least parametric assumption of normality in univariate analyses, principal axis factor analysis was utilised for dimension reduction [43]. Reduction was carried out using a direct obliminal rotation with a delta of 0, given the likelihood of correlations between dimensions of hangover, and in line with the recommendations of Costello and Osborne [43]. Factors with an eigenvalue above 1 were retained, with results checked visually using the Scree test. Results indicated a solution with 2 dimensions, one factor consisting of symptoms linked to dehydration, and one to stress responses. Kaiser–Meyer–Olkin statistics indicated good sampling adequacy (KMO = 0.722), though KMO values for individual variables indicated a potential issue with the item ‘nausea’ (KMO = 0.495; removal of this item did not result in changes to the factor structure). Bartlett’s test of sphericity indicated acceptable deviance from an identity matrix (× 2(28) = 138.762, *p* < 0.001), and 35% of residuals between observed and reproduced correlations had absolute values above 0.05, indicating moderate model fit. The factor model is summarized in Table 3.

Composite scores were calculated for each factor based on the mean of the items which had their primary loadings on each factor (Headache and thirst: mean = 5.32, SD = 1.17; Gastric and cardiovascular symptoms: mean = 3.20, SD = 1.53) with higher scores indicating greater severity of symptoms within the cluster. The bold indicates most relevance.

### 3.2. Regression Models

Three initial regression models of the AHS score were formed, with PCS score and age used as predictor variables in all three models, and the contribution of ‘measures of drinking’ assessed across separate models. A further two regression models were formed to assess dimensions of the AHS identified in factor analysis. Descriptive statistics for the variables included across these regression models are presented in Table 4. To minimize overfitting of the data, rather than utilize an automated variable selection method, for each model, each variable was entered into the regression concurrently [44], with a model formed for each of the three ‘measures of drinking’ obtained; units consumed, eBAC, and eBAC15%. Gender was also entered into the model with the number of units consumed, since this is controlled for in the calculations used to derive eBAC and eBAC15% scores and approximates differences in the body fat composition of different genders which influences alcohol distribution through body water during consumption.

Results indicated that eBAC represented the drinking measure that explained the most variance in AHS scores (power = 0.50, calculated post-hoc), and as such the model containing this variable was carried forward for regression analyses of factor scores calculated during factor axis analysis. A regression model containing eBAC, PCS score, and age, as predictor variables, was therefore formed for each of the two factor scores derived. Summaries of regression models are presented in Table 5, with a summary of individual variable contributions presented in Table 6.

Tolerance, variance inflation factors (VIFs), and collinearity diagnostics indicated no issues with multicollinearity. Manual examination of standardized residuals plotted against standardized predicted values suggested no issues with heteroscedasticity in the data, and multivariate normality was present in all models. Durbin–Watson statistics indicated independence of errors. Some issues were identified in casewise diagnostics, with a small number of cases indicating issues with either problematic covariance ratios or high leverage values in regression models.

Given that a number of cases presented potential issues with covariance and leverage, and in line with recommendations made by Babyak [44], validation of the final models (those containing the eBAC drinking measure) was carried out using bootstrap methods with 2000 random resamples drawn. Bootstrapped models are summarised in Table 7.

Results of bootstrap analyses have a fairly high level of agreement with original regression models, with significant predictor variables remaining constant. Both eBAC and total PCS score demonstrated significant relationships with total AHS score, and a composite score based on symptoms of the AHS related to ’Headache and thirst’. Only total PCS score demonstrated a relationship with a composite score based on ‘Gastric and cardiovascular’ symptoms. eBAC approached significance in this model. However, bootstrapping did indicate some bias toward significance of this variable with this sample.

## 4. Discussion

### 4.1. Summary of the Main Findings

Hangover, the mental and physical symptoms experienced the day after drinking and once BAC is approaching 0, has previously been associated with a variety of other factors, including genetic influences on alcohol metabolism [7,8], gender [9], inflammatory responses [10,11], immunological function [12] and congeners [13]. However, few psychosocial predictors of hangover have been identified so far.

The three main aims of the current study were: (i) to examine whether increased PC scores are associated with elevated hangover severity scores, (ii) to explore the factor structure of the acute hangover scale (AHS), and (iii) to explore whether different dimensions of the AHS were independently associated with PC. The current study demonstrated that PC was a predictor of perceived hangover severity and was, in fact, a stronger predictor than the estimated peak blood alcohol concentration (eBAC). Exploration of the dimensions of the AHS revealed two distinct symptom dimensions; ‘Headache and thirst’; and ‘Gastric and cardiovascular’ symptoms. While both eBAC and PC were significantly associated with ‘Headache and thirst’, only PC was associated with ‘Gastric and cardiovascular’ symptoms.

### 4.2. Relationships between Pain Catastrophising and Hangover Severity

Relationships between PC and AHS scores were investigated using multiple linear regression. Initial models indicated that, of the drinking measures, a calculated blood alcohol concentration that did not account for elimination (eBAC) was the best predictor of the AHS score. One potential explanation is that the preferential metabolism of ethanol limits downstream action to eliminate ethanol metabolites leading to a build-up of biologically active compounds [45]. This would be consistent with the time course of alcohol hangover, with symptomology extending beyond the period of acute ethanol intoxication.

Regression models (and calculated product measures [46]) indicated PC was a better predictor of perceived hangover severity than eBAC. Given relationships between PC and other psychosocial variables such as depression and anxiety [20,21], PC could provide a mechanism through which other psychosocial variables influence self-report hangover severity scores. Furthermore, given links between PC and inhibitive processes [26], this cognitive strategy may influence motivational responses to hangover, providing a potential link between hangover experience and local behaviour, such as engagement with further drinking. Such effects could have implications in addiction research [15].

The results of the current investigation may also have implications for the measurement of hangover, given its reliance on self-report measures, such as the AHS [32]. This issue has been largely ignored in hangover research for purposes of practicality, with a lack of other approaches available. Results from the regression models developed as part of this investigation indicate a moderate effect of PC on AHS score, comparable to the effect observed for measures of alcohol consumption, and support the view that self-report hangover questionnaires contain a significant subjective element. This may reinforce the need for an objective measure of hangover. However, research into biomarkers of hangover severity has yet to find a reliable indicator [1,28]. An alternative approach to measuring hangover severity in a more objective manner may be to examine the cognitive effects of hangover. A meta-analytic examination of the next-day cognitive effects of hangover published in 2018 suggested that effects can be seen during hangover on short- and long-term memory, sustained attention, and psychomotor speed [47]. Differences in performance on tasks examining these functions between hangover and non-hangover days could therefore present a measure of functional hangover severity.

Questions can be raised regarding the value of any of these measurement approaches. Arguably, the subjective experience of hangover is likely to influence the behavioural response to the experience, and may provide value over ‘objective’ measurements of hangover, such as cognitive performance measures or biomarkers, in particular contexts (e.g., the investigation of absenteeism/presenteeism and other acute behaviours). In comparison, objective measures may be more useful in investigations examining the biological correlates of alcohol hangover. Further research will need to examine the comparative value of different measurement approaches in relation to different outcomes. However, controlling for PC in future analyses may also aid in understanding the hangover experience, particularly with regard to the investigation of biomarkers.

#### Dimensions of the AHS

A recent review of the physiology of hangover identified alcohol metabolites, neurotransmitter alterations, inflammatory factors, and mitochondrial (metabolic) dysfunction as the most likely factors involved in hangover symptomology [48]. PC has also been associated with alterations to immune responses, with heightened reactivity of cytokine IL-6 related to increased levels of PC as measured immediately after painful stimulation [49]. This relationship between PC and IL-6 also appeared to be independent of pain ratings given during stimulation. Likewise, immune responses during hangover have been shown to include increases in IL-6 levels [28], with IL-6 thought to have particular importance as a messenger molecule that connects peripheral regulatory processes with the central nervous system during responses to both physiological and psychological stress [49].

In this study, factor analysis of AHS responses resulted in two symptom dimensions; (1) Headache and thirst (‘headache’, ‘tired’, ‘thirsty’), and (2) Gastric and cardiovascular symptoms (‘nausea’, ‘dizziness/faintness’, ‘heart racing’, ‘loss of appetite’, and ‘stomach ache’). The ‘Headache and thirst’ symptom cluster could be related to the diuretic properties of alcohol [29], which can lead to dehydration. Dehydration has been linked to headache [50], with tiredness and thirst being considered common symptoms. Headache may also be the result of cytokine release prompted by physiological stress associated with alcohol consumption [29,30], or indeed physiological stress may be caused by dehydration. However, there is potential for overlap in the causes of symptom clusters. Speculation regarding the biological mechanisms underlying symptom cluster experience is, however, not possible based on the current investigation. Future work will be needed to identify specific biological associations with the experience of hangover symptom clusters. Penning et al.’s factor analysis also identified dehydration (‘disturbed water balance’) as a dimension of the hangover experience [2]. However, in their investigation, the item ‘headache’ was not loaded on this dimension. Dehydration causes physiological changes, e.g., to electrolytic balance, which have proposed associations with hangover. However, evidence for relationships between physiological changes and hangover severity is lacking [12], though they have not been investigated in relation to specific symptom clusters. One potential explanation for the ‘Gastric and cardiovascular’ symptom cluster emerging is that they can all be linked to physiological stress responses. Effects of stress response on the autonomic nervous system are well established [51], and acute physiological stress can also induce various responses in the gastrointestinal system [52].

An alternative explanation of the factor structure of the AHS identified in this investigation relates to the prevalence of symptoms. Tiredness, thirst and headache, the items loaded within the ‘Headache and thirst’ dimension, represent the three most commonly reported hangover symptoms [31]. It is possible that these symptom clusters are thus representative of different groups that either experience one or both of the symptom clusters. An extension of this reasoning, given the prevalence of headache and thirst symptoms in hangover, could be that less severe hangovers consist of symptoms included within the ’Headache and thirst’ symptom cluster, with more severe hangovers including ‘Gastric and cardiovascular’ symptoms.

### 4.3. Dimensions of the AHS Independently Associated with PC

Composite scores based on ‘Headache and thirst’ symptoms, and ‘Gastric and cardiovascular’ symptoms, identified during factor analysis of AHS responses, were also assessed using regression. Both eBAC and PCS score significantly predicted ‘Headache and thirst’ symptom scores with approximately equal contributions. The observation of PC score as a significant predictor in this model is possibly due to the inclusion of headache severity ratings in the construction of this score, with PC having previously been linked with both the presence of weekly headache [53], and the severity of migraine symptoms, a phenomenon associated with headache [54]. Given the diuretic effects of alcohol [33], it follows that measures of alcohol consumption would be related to symptoms associated with dehydration.

Finally, only PCS score significantly predicted composite scores based on ‘Gastric and cardiovascular’ symptoms, though eBAC was only marginally non-significant. Product measures supported the interpretation that PC was more strongly related with ‘Gastric and cardiovascular’ symptoms than eBAC, and robust regression provided some validation of this model. PC has been related to activity in the mPFC [24], an anatomical area that has also been shown to mediate stress response [55]. This may provide a link through which this cognitive strategy can influence stress responses occurring as a result of hangover. The exclusion of eBAC from this model may indicate that these symptoms are not direct products of alcohol consumption, or that this symptom set is not associated linearly with the volume of alcohol consumed (e.g., threshold effects). It has, however, been previously suggested that increased levels of fatty acids seen during hangover are products of a stress response concurrent with hangover [12], which could indicate that ‘Gastric and cardiovascular’ symptoms in hangover are somewhat independent of the amount of alcohol consumed.

### 4.4. Conclusions and Directions for Future Research

Hangover represents a considerable economic toll due to its influence on local behaviour, such as lost productivity and workplace absenteeism [4]. Furthermore, the experience of hangover may be related to downstream health consequences by promoting deviant drinking practices [12]. The current investigation revealed, for the first time, that PC predicts alcohol hangover severity and that this effect occurs in a symptom specific manner. PC may also provide a cognitive strategy through which other psychosocial variables can influence hangover.

Exploratory factor analysis provided evidence of two distinct sub-structures of the AHS, ‘Headache and thirst’, and ‘Gastric and cardiovascular’ symptoms. Results of this investigation could be interpreted as suggesting that dehydration and physiological stress responses represent areas that warrant further examination, with differences in regression models based on composite hangover scores for symptom clusters providing some evidence that symptom clusters are somewhat independent. This may provide an explanation for why markers of dehydration have not always correlated with overall hangover severity [12], as well as why thirst had the lowest item-total correlation during development of the AHS [32]. Further research will be required to establish whether particular covariables correlate with symptom clusters either derived from dimension reduction procedures or theoretical mechanistic relationships. The AHS also measures a somewhat limited sample of hangover symptoms, and recent research has adopted the approach of combining the symptoms identified in a number of validated hangover measures, in order to capture the diversity of the hangover experience [56]. These measures consist largely of different symptoms, but show high correlations, and further research will be needed to examine whether the dimensions of the hangover experience suggested here are evident within this broader context, as well as their relationships with PC.

As noted previously, hangover has also been associated with effects on local drinking behaviour [17,18], with an ecological momentary assessment conducted by Epler et al. in 2014 indicating that the presence of hangover delayed the onset of the next drinking episode when interacting with either the onset of financial stressors, or the presence of craving at the end of the drinking episode [18]. Epler et al.’s (2014) sample consisted of participants with a reasonably low risk of alcohol problems (average AUDIT score = 12.21), but no research has addressed this relationship in high-risk or clinical groups. Evidence has suggested relationships between craving and AUD symptomology in a sample containing a high proportion of participants meeting criteria for diagnosis of AUD. However, no relationship was found between craving and drinking habits in this sample [57]. This may suggest that interactions between craving, hangover, and local drinking behaviour do not exist in those at a high risk for AUD. Greater craving in the high-risk sample also showed a relationship with increased impulsive discounting (a devaluation of future reward) [57], which may provide a mechanism for observed losses of inhibitory response control in alcohol disorders, as well as other disorders, such as depression [58]. Weaker inhibition processes have also been noted in those with a family history of AUD [59], and in young-adult binge drinkers [60], a form of drinking associated with an increased incidence of hangover. Inhibition is also inherently linked with impulsivity [61], which has itself been strongly associated with AUD [58]. Given the links between PC and motivational/inhibitive processes [26,62], future research should consider PC and hangover alongside factors related to motivation/inhibition, such as performance on inhibition dependent tasks, and craving. Vatsalya et al.’s (2018) investigation found no relationship between hangover severity (as measured by the AHS) and a single item measure of craving [40]. However, this craving measurement is unlikely to capture the theoretical complexity of the phenomenon and future research would benefit from the use of context appropriate, validated craving measures [63].

Future research should therefore seek to elucidate the potential interaction between PC and cognitive processing systems mediating inhibitory control and the craving response during alcohol hangover.

## Figures and Tables

**Table 1 jcm-09-00280-t001:** Descriptive statistics for ratings of hangover symptom severity on the acute hangover scale (AHS).

Item.	Mean	SD	Median	Normality	95% CI	
					Lower	Upper
Tired	5.94	1.24	6	<0.001 ***	5.68	6.21
Thirsty	5.31	1.528	5	0.001 ****	4.99	5.64
Headache	4.71	2.057	5	<0.001 ****	4.27	5.15
Nausea	3.66	2.433	3	<0.001 ****	3.14	4.18
Loss of appetite	3.33	2.078	3	<0.001 ****	2.88	3.77
Dizziness/faintness	3.22	2.008	3	<0.001 ****	2.79	3.65
Stomach ache	3.06	2.054	2	<0.001 ****	2.62	3.5
Heart racing	2.72	2.096	2	<0.001 ****	2.27	3.17

SD—standard deviation; normality—*p*-Value for Shapiro–Wilk analysis of normality. The *n* for all items was 86. Significant results indicated by *** *p* < 0.01, **** *p* < 0.001.

**Table 2 jcm-09-00280-t002:** Correlations (and significance) of items included in factor analysis of AHS.

	Thirsty	Headache	Nausea	Loss of Appetite	Dizziness/Faintness	Stomach Ache	Heart Racing
Tired	0.239	0.312	0.192	0.176	0.156	0.223	0.161
	(0.013) *	(0.002) ***	(0.038) *	(0.052)	(0.075)	(0.020) *	(0.069)
Thirsty		0.276	0.140	0.112	−0.031	0.084	0.226
		(0.005) **	(0.100)	(0.152)	(0.390)	(0.221)	(0.018) *
Headache			0.208	0.344	0.241	0.182	0.183
			(0.027) *	(0.001) ***	(0.013) *	(0.047) *	(0.046) *
Nausea				0.473	0.497	0.529	0.434
				(<0.001) *	(<0.001) ****	(<0.001) ****	(<0.001) ****
Loss of appetite					0.369	0.211	0.305
					(<0.001) *	(0.026) *	(0.002) *
Dizziness/faintness						0.251	0.409
						(0.010) *	(<0.001) ****
Stomach ache							0.362
							(<0.001) ****

Significant correlations indicated by * *p* < 0.05, ** *p* < 0.01, *** *p* < 0.005, **** *p* < 0.001.

**Table 3 jcm-09-00280-t003:** Factor loadings of AHS items based on principal axis factoring.

Item	Headache and Thirst	Gastric and Cardiovascular Symptoms
Headache	**0.587**	0.082
Thirsty	**0.506**	−0.069
Tired	**0.456**	0.072
Nausea	−0.102	**0.894**
Dizziness/faintness	−0.065	**0.641**
Heart racing	0.088	**0.535**
Stomach ache	0.024	**0.532**
Loss of appetite	0.164	**0.47**
Eigenvalues	1.252	2.906
Factor correlation	0.452	

**Table 4 jcm-09-00280-t004:** Descriptive statistics for variables included across the five regression models constructed.

Variable	Mean	SD	Median	Normality	95% CI	
					Lower	Upper
Acute Hangover Scale (AHS)	4.1	1.2	4.11	0.101	3.85	4.36
Pain Catastrophizing scale (PCS)	28.81	11.64	26.5	<0.001 ****	26.32	31.31
Age	25.93	6.03	25	<0.001 ****	24.64	27.22
Total units	15.62	8.64	13.4	<0.001 ****	13.77	17.48
eBAC	0.26	0.14	0.22	<0.001****	0.23	0.29
eBAC15%	0.18	0.13	0.14	<0.001 ****	0.15	0.21
Headache and thirst	5.32	1.17	5.33	0.06	5.07	5.57
Gastric and cardiovascular symptoms	3.2	1.53	3	0.002 ***	2.87	3.53

Variables: AHS—acute hangover scale; PCS—pain catastrophising scale; total units—units of alcohol consumed, calculated from self-report; eBAC—the estimated blood alcohol concentration assuming no elimination; eBAC15%—the estimated blood alcohol concentration assuming 15% elimination rate; Headache and thirst—mean score for items identified within dimension 1 of the factor analysis; Gastric and cardiovascular symptoms—mean score for items identified within dimension 2 of the factor analysis; SD—standard deviation; normality—significance of Shapiro–Wilk analysis. The *n* for all measures was 86, *** *p* < 0.005, **** *p* < 0.001.

**Table 5 jcm-09-00280-t005:** Summary of model statistics for regression analyses.

Model	DV	IV	R	R^2^	Adj R^2^	F	F Sig.	Durbin–Watson
1	Acute Hangover Scale (AHS)	Units consumed	0.432	0.187	0.147	4.656	0.002 ***	1.789
		Gender						
		PCS score						
		Age						
2	AHS	eBAC15	0.397	0.158	0.127	5.118	0.003 ***	1.771
		PCS score						
		Age						
3	AHS	eBAC	0.429	0.184	0.154	6.163	0.001 ***	1.802
		PCS score						
		Age						
4	Headache and thirst	eBAC	0.307	0.094	0.061	2.844	0.043 *	1.612
		PCS score						
		Age						
5	Gastric and cardiovascular symptoms	eBAC	0.398	0.158	0.128	5.141	0.003 ***	1.906
		PCS score						
		Age						

DV (dependent variable): AHS—Acute hangover scale; Headache and thirst—mean score for items identified within dimension 1 of the factor analysis; Gastric and cardiovascular symptoms—mean score for items identified within dimension 2 of the factor analysis. R—value of r for model; R2—value of r squared for model; Adj R2—adjusted r squared for model; F—F value for model; F Sig.—Significance of the F value for the model; Durbin-Watson—Durbin-Watson statistic for the model. Significant results indicated by *, * *p* < 0.05, *** *p* < 0.005.

**Table 6 jcm-09-00280-t006:** Summary of statistics determining independent variable contributions to regression effects.

Model	DV	IV	B	SE B	β	t	t Sig.	95% Confidence Intervals		Correlations			Tolerance	VIF	Pratt
								Lower Bound	Upper Bound	Zero-Order	Partial	Part			
1	Acute Hangover Scale (AHS) score	Constant	2.955	0.709		4.17	<0.001 ****	1.545	4.365						
		Units consumed	0.033	0.014	0.239	2.342	0.022 *	0.005	0.061	0.175	0.252	0.235	0.967	1.034	0.042
		Gender	0.366	0.25	0.151	1.46	0.148	−0.133	0.864	0.186	0.16	0.146	0.939	1.065	0.028
		PCS score	0.032	0.011	0.314	3.038	0.003 ***	0.011	0.053	0.326	0.32	0.304	0.94	1.063	0.102
		Age	−0.02	0.02	−0.1	−0.991	0.324	−0.06	0.02	−0.148	−0.109	−0.099	0.981	1.019	0.015
2	AHS score	Constant	3.255	0.695		4.682	<0.001 ****	1.872	4.637						
		eBAC15	1.867	0.954	0.2	1.957	0.054	−0.03	3.765	0.209	0.211	0.198	0.982	1.018	0.042
		PCS score	0.033	0.011	0.317	3.106	0.003***	0.012	0.054	0.326	0.324	0.315	0.985	1.015	0.103
		Age	−0.017	0.02	−0.083	−0.809	0.421	−0.057	0.024	−0.148	−0.089	−0.082	0.968	1.033	0.012
3	AHS score	Constant	3.005	0.7		4.292	<0.001 ****	1.612	4.398						
		eBAC	2.262	0.881	0.257	2.568	0.012 *	0.51	4.014	0.26	0.273	0.256	0.99	1.01	0.067
		PCS score	0.033	0.01	0.321	3.189	0.002 ***	0.012	0.054	0.326	0.332	0.318	0.984	1.016	0.105
		Age	−0.017	0.02	−0.085	−0.839	0.404	−0.057	0.023	−0.148	−0.092	−0.084	0.976	1.025	0.013
4	Headache and thirst	Constant	4.288	0.721		5.943	<0.001 ****	2.852	5.723						
		eBAC	1.876	0.907	0.218	2.067	0.042 *	0.07	3.681	0.216	0.223	0.217	0.99	1.01	0.047
		PCS score	0.022	0.011	0.216	2.039	0.045 *	0.001	0.043	0.214	0.22	0.214	0.984	1.016	0.046
		Age	−0.003	0.021	−0.015	−0.141	0.888	−0.044	0.038	−0.062	−0.016	−0.015	0.976	1.025	0.001
5	Gastric and cardiovascular symptoms	Constant	2.323	0.908		2.559	0.012 *	0.517	4.129						
		eBAC	2.258	1.142	0.201	1.978	0.051	−0.013	4.529	0.208	0.213	0.2	0.99	1.01	0.042
		PCS score	0.039	0.013	0.299	2.931	0.004 ***	0.013	0.066	0.311	0.308	0.297	0.976	1.025	0.093
		Age	−0.032	0.026	−0.128	−1.245	0.217	−0.084	0.019	−0.183	−0.136	−0.126	0.976	1.025	0.023

DV (dependent variable): AHS—Acute hangover scale; Headache and thirst—mean score for items identified within dimension 1 of the factor analysis; Gastric and cardiovascular symptoms—mean score for items identified within dimension 2 of the factor analysis. B—Beta coefficient; SE B—Standard error of beta coefficient; β—standardized beta coefficient; t—t-statistic value for parameter; t Sig.—significance of t-statistic for parameter; VIF—Variance inflation factor; Pratt—Pratt statistic for parameter. Significant results indicated by *, * *p* < 0.05, *** *p* < 0.005, **** *p* < 0.001.

**Table 7 jcm-09-00280-t007:** Summary of bootstrapped regression model coefficients.

Model	DV	IV	B	Bias	SE	Sig. (Two-Tailed)	BCa 95% CI	
							Lower Bound	Upper Bound
3	Acute Hangover Scale (AHS) score	Constant	3.005	0.009	0.725	<0.001 ****	1.612	4.491
		eBAC	2.262	0.027	0.959	0.022 *	0.483	4.280
		PCS score	0.033	<0.001	0.009	<0.001 ****	0.015	0.051
		Age	−0.017	<0.001	0.020	0.401	−0.058	0.021
4	Headache and thirst	Constant	4.288	0.046	0.658	<0.001 ****	2.975	5.840
		eBAC	1.876	0.032	0.950	0.046 *	0.049	3.812
		PCS score	0.022	<0.001	0.010	0.026 *	0.003	0.040
		Age	−0.003	−0.002	0.022	0.890	−0.047	0.035
5	Gastric and cardiovascular symptoms	Constant	2.323	−0.002	0.922	0.014 *	0.577	4.119
		eBAC	2.258	0.007	1.233	0.066	−0.060	4.654
		PCS score	0.039	<0.001	0.013	0.004 ***	0.013	0.064
		Age	−0.032	<0.001	0.024	0.165	−0.080	0.015

DV—dependent variable; IV—independent variable; B—beta weight; SE—standard error; BCa 95% CI—bias-corrected accelerated 95% confidence interval. Bootstrap results based on 2000 bootstrap samples. Significant results indicated by *, * *p* < 0.05 = *, *** *p* < 0.005, **** *p* < 0.001.

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
