# Peer review of "Pain Catastrophising Predicts Alcohol Hangover Severity and Symptoms"

_jcm, 2020, doi:10.3390/jcm9010280_

Round 1

Reviewer 1 Report

Sufficient changes have been made to the document.

Reviewer 2 Report

The response is satisfactory. The manuscript is now acceptable. 

This manuscript is a resubmission of an earlier submission. The following is a list of the peer review reports and author responses from that submission.

Round 1

Reviewer 1 Report

            This study is an interesting and novel investigation into the role that the cognitive style involved in pain catastrophizing may play in the severity of self-reported hangover.  Few psychosocial predictors of hangover have been found so far, and this cognitive style is predictive of hangover symptom reports. As such, it makes a novel and interesting contribution.

            The strongest concern is how the factors of the Acute Hangover Scale are labelled.  Interpreting the meaning of factors or components is an art, not a science, and involves some judgment calls in identifying the core concept. Factors should be labelled in a manner that better conveys the content of the factor, not an interpretation of what may or may not underlie the cluster of symptoms. In this case, the anchor item for one factor is “headache” and for the other factor is “nausea”. Apparently a decision was made to interpret headache and tiredness as being due to dehydration, although the headache may result directly from cytokine release rather than dehydration itself. It would be more accurate to call that factor “Headache/thirst” or “Pain/thirst”, to clarify what the two highest rated elements are. A more serious concern is referring to the rest of the items as “stress-related” since that seems far-fetched. Three of the items, including the anchor “nausea” are related to gastric distress, which is not likely to be caused by stress in this case, and dizziness/faintness is also unlikely to be a stress response.  While stress can cause the heart to race, a great many physical conditions can cause heart rate to increase (including nausea) without involving stress as a causative factor. It would be more accurate to tie the label more closely to the items, such as “Gastric distress and dizziness”.  Speculation about what underlies the symptom clusters could be saved for the Discussion. It would then make more sense that pain catastrophizing is related to “Pain/Thirst” items and to how strongly people report gastric distress.

            First sentence of Intro: tremors are not validated as a sign of hangover, despite what Penning said that one time. The Verster et al. (2010) consensus paper from the international Alcohol Hangover Research Group provides a list of symptoms that have been validated in controlled investigations in its second sentence. Yhilikari et al (1974) distinguished signs (observable) from symptoms (subjective) in a controlled investigation and found no support for signs such as tremor. Some researchers mistakenly believe that hangover represents mild withdrawal (and thus list withdrawal symptoms) but withdrawal and hangover involve different methods of induction and CNS systems (Prat et al (2009). E.g., Slutske’s scale includes withdrawal symptoms, not just validated hangover symptoms.

            Second paragraph, better to remove “such as guilt” (an odd choice for a variable that might cause hangover) to make that sentence broader in scope. The next paragraph has a much more informed list of possibilities.

In Section 1.2, when talking about time to next drink, it would be useful to add reference to Huntley et al (2015) who showed that on drinking days, more severe hangover predicted a lower quantity of drinking later than day (n = 274).

In making the case for the involvement of pain in hangover, it would be worthwhile including the limited evidence on the role of cytokines (Kim et al, 2003), since increasing attention is being paid to that process (Kim’s study gets cited repeatedly in other articles on the pathophysiology of hangover). This may provide a solid physiological basis for the pain of headache following heavy drinking.

            Table 1: Put the items in order of mean rating, to make it easier to navigate. Table 2: put the items in order of factor loadings on the first factor, then on the second factor.

            The Methods needs to add the rating scales and method of scoring the AHS and the PCS. (E.g., informing readers that the AHS involves rating each item from 0 “None” to 7 “Incapacitating”, with the mean of the ratings scored.)

            Regressions: Any variables that are collinear (r > .90) should not be entered into the same regression – it causes wildly unstable results and suppressor effects. I would suspect that the two eBAC variables are collinear. Check for collinearity among variables entered into the same regression. Since gender effects in the eBAC variables are already controlled for, that is not a reason to include gender.  Come up with a more theoretical reason to include it, or else exclude it.

            The list of the variables entered into the first regression or regressions is not described but needs to be clear at the start of 3.2. If all variables from the Table 3 were entered together, this is invalid since the AHS total is a linear combination of its two factors. Only the two factors should be entered.  Either that or only use the total score without the factor scores, but that would be less interesting. Table 4 is confusing: Are there five separate regressions or are those five steps of one regression? Put that information in the table title.  In any case, it is too hard to determine what is in each regression (or step) the way the table is laid out. Indicate more clearly if the statistics in columns 4-8 are for the entire model (or step) as opposed to belonging to the variable to the immediate left of those values.  When R2 is given, R is not also needed.  Look at examples from other journal articles about how to more clearly present regressions in tables.

            Consider presenting a table of the correlations of the AHS items (grouped by factor) to make it easier to see shared variance. This is important information when considering the shared variance between thirst and the other two items in that factor.

            The Discussion has a far too narrow interpretation of the possible causes of the hangover symptoms.  Even if stress can sometimes cause certain symptoms, most of these are far more commonly caused by other physical processes so it is overly presumptuous to say that after a night of heavy drinking, it is feeling stressed, not physiological reactions to alcohol per se, that cause nausea, etc. It would make more sense to consider how heavy drinking could affect physiological processes that lead to gastric distress and dizziness, as other scholars have done. Similarly, other possible causes of the headache and tiredness need to be considered. Scholarly reviews of underlying pathophysiology of hangover are published and can be referred to.

            There is no point in referring to the “hair of the dog” theory (section 4.4), given that only 11% of social drinkers did that at all in the past year, and the effectiveness of that approach was reported to be minimal (survey by Verster et al 2010).

            When saying that Penning showed that thirst does not always correlate with hangover severity, it would be worth adding that in the development of the AHS, thirst had the lowest item-total correlation while nauseas/dizziness had the highest.

Reviewer 2 Report

The manuscript submitted by Royle et al. provides new and fascinating data on the functional relationship between personality traits (i.e., the tendency for pain catastrophizing / PC) and subjective hangover ratings, which are often used as the main way of assessing the severity of an alcohol hangover.

Study motivation, design and analyses are sound. The provided conclusions are justified and genuinely new.

Yet, the manuscript would benefit from a revision of the line of arguments, where the authors correct some inaccuracies and focus more on what has actually been investigated in the current sample. As all of these issues may easily be addressed in a revision, the authors have to addressed the issues enlisted below.

1. Please rectify the definition of “cognitive control”:

In the abstract, the authors write that PC, i.e., rumination and exaggeration are “cognitive control processes”. Yet, the PCS treats PC rather like a personality trait, albeit situation-bound, than like a skill or cognitive faculty. While I do agree that giving in to rumination and exaggeration could potentially be interpreted as a lapse of (inhibitory) control, I would certainly not deem PC itself to represent a control process (compare Diamond, 2013, doi: 10.1146/annurev-psych-113011-143750 or Miyake et al., 2000, doi: 0.1006/cogp.1999.0734) and would also suspect that aside from control, affective aspects should also play a role in PC. Lastly, it should be noted that DNIC/CPM is certainly not the same as cognitive or behavioral top-down control processes and that, albeit some anatomical overlap, there is not clear functional link between motor response inhibition and the cognitive evaluation of pain (as claimed in paragraphs 1.3, 4.2, and 4.4). Please correct this in the entire manuscript.

2. Please limit the research goals to what was actually investigated:

A. ABSTRACT: The authors state that underlying processes “influence experience of alcohol hangover which may be important factors mediating the motivation to drink and/or abuse alcohol.” Yet, the subsequent research questions / study goals do not explicitly mention the investigation of drinking motivation or incidents, if I understood it correctly. Please either include such a research question plus adequate analyses in the abstract and in the manuscript, or delete it from that section (it can optionally still be shifted to the discussion & outlook section of the abstract).

B. INTRODUCTION: Please try to avoid evoking the impression that you deem PC to be a “biological mechanism(s) that underpin(s) the experience of alcohol hangover” (lines 47/48) when the main analyses were correlations of different questionnaire-derived measures and score. -The way it was assessed, PC is not an objective lab-based parameter reflecting a clearly identifiably biological/neuronal mechanism or correlate. This also accounts for lines 317-318 and 4.4.

C. INTRODUCTION: Given that the authors neither assessed participants with AUD, nor alcohol-related conditioning, nor the risk of developing AUD, the entire paragraph 1.2 seems quite superfluous to me. As none of this was investigated in the given study, please consider to delete, or at least substantially shorten, the entire paragraph. In order to justify the study motivation, I would deem sufficient to go with a more straight-forward line or arguments (i.e., ~Hangover severity is commonly assessed with questionnaires, while it has been largely ignored that such ratings may be heavily modulated by several factors that might affect subjective ratings. Given that some of the common hangover symptoms relate to pain, PC is a trait that may affect hangover ratings and thereby bias study results. This shall hence be investigated.)

Also, there is a functional misunderstanding in lines 55-58 of that paragraph (“there is some evidence that alcohol hangover experience is a risk factor for alcohol use disorder (AUD) [16]. In this regard hangover could be conceptualized as affecting cognitive control processes that influence the motivation to drink by conditioning”): -The cited reference by Piasecki et al. [16] does not allow for the oversimplifying statement that hangover is a clear risk factor for AUD. More importantly, however, cognitive control processes are not affected by (AUD-related) conditioning, as conditioning alone does not diminish cognitive control capacities. Conditioned behavior may however “raise the bar” by requiring more and more control capacities, as dysfunctional behaviors are increasingly reinforced and S-R associations are established (compare Norman & Shallice, 1986 or Stock 2017, doi: 10.3389/fpsyg.2017.00884).

3. Please define alcohol hangover more precisely:

Toxicologically speaking, alcohol hangover does likely not start at a fixed time since the last drink. Depending on physical health and on how much has been consumed, sobriety may or may not be reached after 6 to 8 hours, and any presence of (residual) ethanol in the system should be rated as intoxication. Hangover should therefore be best defined by reaching a BAC of zero, as well as by the presence of adverse drinking-related symptoms (e.g., van Schrojenstein Lantman et al., 2016, doi: 10.2174/1874473710666170216125822)

4. Please relate the AHS questionnaire to recent findings:

For the sake of comparison and completeness, please briefly describe the differences between the AHS questionnaires and other, recently applied questionnaires like those of van Schrojenstein Lantman, especially van Schrojenstein Lantmann et al. (2017, doi: 10.1002/hup.2623), who used quite a few comparable items .

5. Please provide additional analyses:

Given that the PCS specifically assesses the evaluation of pain, I could not help but wonder what would happen if a score made up of the pain related items/symptoms only (i.e., stomach ache and head ache) was compared to a composite of all the other symptoms (i.e., thirsty, tired, dizzy, loss of appetite, nausea, heart racing).

6. Please provide an extended outlook:

A. In the discussion, please (again) give a quick overview / list of other factors which have also been demonstrated to modulate hangover severity (like sleep, congeners, immune fitness etc.) to provide the reader with a more complete and exhaustive overview of hangover-modulating factors, and embed the current findings in this context.

B. Given that PC may bias (subjective) hangover ratings quite heavily, I would encourage to discuss the implications for hangover research a bit more openly and critically: Based on your findings, what are your opinions on using subjective hangover ratings as (sole) measures in hangover research? Should PC be used as a covariate in future studies? Should more objective measures, like ETG, EtS, histamine, vasopressin, or cytokines be used to complement subjective ratings and compensate for biases like PC - if so, which and why?

MINOR:

In the tables, please mark significant effects with an asterisk next to the p values Please note that by policy of the journal, all data and code must be provided in an openly accessible repository (https://www.mdpi.com/journal/jcm/instructions#suppmaterials). Please remember to provide a link and upload your data as part of the review.

Reviewer 3 Report

A very interesting paper with just a few points to consider and minor amendments to make.

36: I think there may be a better way to introduce the Alcohol Hangover e.g. see The van Schrojenstein Lantman's (2017)  definition as Prat, Adan and Sánchez-Turet (2009) only suggest 6-8 hours after ingestion.

40: Insert 'a' between 'that' and 'better understanding'.

73: Sentence starting with 'Given' is somewhat long winded and could be restructured.

133: Spell out number in words when starting a sentence or restructure.

155: Unsure what items you refer to from McKinney and Coyle's study regarding consumption prior to most recent hangover. Please specify. Also, McKinney 'and' Coyle rather than '&' outside brackets.

285: Full stop after 'et al'.